# The Acute and Chronic Effects of Resistance and Aerobic Exercise in Hemostatic Balance: A Brief Review

**DOI:** 10.3390/sports11040074

**Published:** 2023-03-27

**Authors:** Apostolos Z. Skouras, Dimitrios Antonakis-Karamintzas, Andreas G. Tsantes, Athanasios Triantafyllou, Georgios Papagiannis, Charilaos Tsolakis, Panagiotis Koulouvaris

**Affiliations:** 1Sports Excellence, 1st Department of Orthopaedic Surgery, School of Medicine, National and Kapodistrian University of Athens, 12462 Athens, Greece; apostolis.sk@gmail.com (A.Z.S.);; 2Laboratory of Haematology and Blood Bank Unit, “Attiko” Hospital, School of Medicine, National and Kapodistrian University of Athens, 12462 Athens, Greece; 3Biomechanics Laboratory, Department of Physiotherapy, University of the Peloponnese, 23100 Sparta, Greece; 4Sports Performance Laboratory, School of Physical Education & Sports Science, National and Kapodistrian University of Athens, 17237 Athens, Greece

**Keywords:** hemostasis, coagulation, fibrinolysis, platelets, exercise, training, physical activity, high-intensity interval training

## Abstract

Hemostatic balance refers to the dynamic balance between blood clot formation (coagulation), blood clot dissolution (fibrinolysis), anticoagulation, and innate immunity. Although regular habitual exercise may lower the incidence of cardiovascular diseases (CVD) by improving an individual’s hemostatic profile at rest and during exertion, vigorous exercise may increase the risk of sudden cardiac death and venous thromboembolism (VTE). This literature review aims to investigate the hemostatic system’s acute and chronic adaptive responses to different types of exercise in healthy and patient populations. Compared to athletes, sedentary healthy individuals demonstrate similar post-exercise responses in platelet function and coagulatory and fibrinolytic potential. However, hemostatic adaptations of patients with chronic diseases in regular training is a promising field. Despite the increased risk of thrombotic events during an acute bout of vigorous exercise, regular exposure to high-intensity exercise might desensitize exercise-induced platelet aggregation, moderate coagulatory parameters, and up-regulate fibrinolytic potential via increasing tissue plasminogen activator (tPA) and decreasing plasminogen activator inhibitor (PAI-1) response. Future research might focus on combining different types of exercise, manipulating each training characteristic (frequency, intensity, time, and volume), or investigating the minimal exercise dosage required to maintain hemostatic balance, especially in patients with various health conditions.

## 1. Introduction

The preventive and therapeutic benefits of exercise are well established in population-based cohort studies [1,2]. The current evidence suggests that these benefits are independent of the type of exercise. All types of exercise (endurance, resistance, and high-intensity interval training—HIIT) can reduce all-cause and cardiovascular mortality [3,4,5,6], and indeed affect every biological system [2,7,8,9]. In 2019, Jeong et al. [10] claimed that the size effect of regular exercise on mortality reduction rate is greater in individuals with existing cardiovascular disease (CVD) than in a healthy population. Similarly, exercise in people with established impaired glucose tolerance reduces the incidence of diabetes-related complications such as microvascular or cardiovascular complications, and increases life expectancy [11]. Beyond these clinically important observations, vital to human health is also the delicate balance between clotting and bleeding, namely the hemostatic balance [12,13,14,15,16]. Regular habitual exercise may lower the incidence of CVD by improving an individual’s hemostatic profile at rest and during exertion, but vigorous exercise may increase the risk of sudden cardiac death and venous thromboembolism (VTE) in persons with or without underlying vascular disease [17,18,19,20].

Hemostatic balance refers to the dynamic balance between blood clot formation (coagulation), blood clot dissolution (fibrinolysis), anticoagulation, and innate immunity [16,21]. Platelets, blood vessel walls, vascular endothelial cells, and plasma proteins (coagulatory and fibrinolytic factors) are the primary factors in maintaining hemostatic equilibrium [21,22]. Platelets, for example, play a crucial role in hemostatic balance by stimulating thrombin generation and thrombus formation [23,24]. Similarly, tissue plasminogen activator (tPA), which is inhibited by elevated levels of Plasminogen Activator Inhibitor-1 (PAI 1) [25], is the primary catalyzer in the conversion of plasminogen to plasmin, the principal enzyme involved in blood clot dissolution. The vital role of maintaining hemostatic balance is reflected by the correlation of elevated PAI-1 with many diseases and clinical conditions, such as VTE [26], hypertension [27], stroke [28], coronary artery disease (CAD) [29], Alzheimer disease [30], depression [31], type 2 diabetes mellitus (T2DM) [32], and gestational diabetes [33].

Although lack of exercise is considered a major cause of chronic non-communicable diseases [34], physical exertion has been characterized as a double-edged sword for CVD [17,35], endothelial function [36], immune function [37], and arterial thrombosis [38]. There is still controversy on how exercise affects the hemostatic system in healthy individuals, including professional and well-trained athletes, and the effects of exercise on individuals with chronic diseases, including CVD, T2DM, and hypertension. Despite the broad acceptance of the beneficial effect of regular moderate-intensity continuous training (MICT) on the hemostatic profile [38,39,40], the safety and efficacy of single-bout exercise of any type, including regular resistance training and HIIT, still need to be investigated in healthy individuals. However, a considerable amount of evidence suggests that in susceptible people, vigorous physical exercise may abruptly and transiently raise the risk of acute myocardial infarction (MI) and sudden cardiac death [41]. In this article, high-intensity interval exercise (HIIE) refers to a single bout of exercise, while HIIT refers to a regular exercise program. Consequently, this review aims to investigate the acute and chronic adaptive responses of the hemostatic system to different types of exercise in healthy and patient populations.

## 2. Endurance Exercise and Hemostatic Balance

### 2.1. Acute Effects

The hemostatic response of a single bout of endurance exercise has been investigated for many years in athletes, healthy sedentary individuals, and individuals with chronic diseases, mainly CVD. It is well accepted that moderate intensity is not only safe, but also beneficial for platelet function and fibrinolytic potential in both healthy and patient populations [39,42,43,44]. However, there is contradictory evidence on a vigorous single bout of exercise’s effect on hemostatic balance, such as cardiopulmonary exercise testing (CPET) [45,46]. For example, although a single bout of moderate-intensity exercise in overweight and obese women [45] or vigorous exercise in well-trained individuals might not disturb the hemostatic balance [46], strenuous exercise in sedentary individuals may increase the chance of a cardiovascular event [46]. The current literature indicates a protective effect for elite athletes and well-trained individuals of the hemostatic disturbance following vigorous exercise [47].

#### 2.1.1. Healthy Individuals and Athletes

##### Platelet Function

The acute effect of endurance exercise on platelets activity has been tested for many years in healthy humans [48,49,50]. However, the exact influence of an acute bout of exercise on platelet function and aggregation remains unclear [51]. Although endurance exercise increases platelet activation and aggregation in healthy, sedentary people, most findings imply that only vigorous exercise increases thrombotic potential [52,53]. An increase in platelet count and function after vigorous activity may be due to new platelet cells released from the spleen, bone marrow, and pulmonary vasculature [39,54,55,56]. By boosting the release of procoagulant microparticles from platelets, which are rich in phospholipids and coagulation proteins, vigorous exercise also enhances shear-induced thrombin generation [57,58]. Procoagulant response is also proportional to endothelial shear stress and exercise intensity [54,55]. Higher stress destroys vascular endothelium and stimulates Von Willebrand Factor release (vWF). vWF enhances platelet adherence to arterial injury sites and subsequent coagulation [59]. Shear stress combined with an exponential rise in plasma platelet agonists during severe exercise may cause platelet aggregation [55,60,61,62,63]. Hypercoagulable response to acute exercise has several other proposed mechanisms. Another proposed mechanism is based on how catecholamines may mediate exercise-induced platelet activation [64]. Epinephrine activates alpha2-adrenergic receptors, increasing platelet adhesiveness, aggregation, and fibrinogen binding [65,66]. Physiological amounts of epinephrine may not be enough to trigger this pathway, but epinephrine in the presence of ADP activates platelets in vivo [67]. Generally, vigorous exercise leads to greater levels of epinephrine than moderate-intensity exercise [68]. This is in line with previous findings that indicate an increase in platelet function during high-intensity exercise, but not during that of moderate intensity [52,69,70].

##### Coagulation

Acute endurance exercise causes hypercoagulability, as demonstrated by higher levels of coagulation Factor VIII, thrombin–antithrombin III complex (TAT), prothrombin fragments 1 and 2 (F1 + F2), fibrinogen, and fibrinopeptide A, as well as shorter activated partial thromboplastin time (aPTT) and nonactivated partial thromboplastin durations [71,72]. Compared to those who are sedentary, physically trained individuals in response to acute exercise demonstrate decreased vWF antigen and activity, endogenous thrombin potential (ETP), fibrinogen antigen, and F1 + F2, and shortened or equal clotting times (aPTT, PT, and thrombin time—TT) [46].

Other factors which contribute to hemostatic balance are tissue factor pathway inhibitor (TFPI) and tissue factor (TF). TFPI is a procoagulant protein inhibitor that has a time-of-day effect with increased activity during the morning in a rest state [73,74], while the TF, a primary cellular initiator of blood coagulation also demonstrates its highest activity in the morning, but during exercise no time-of-day effect has been observed in these factors [75]. These two seemingly contradictory findings can inform clinicians, sports scientists, and researchers that in the morning hours during rest, fibrinolysis is less active, and during exercise, coagulation is more active, compared to later in the day, predisposing individuals to hypercoagulability and cardiovascular events.

Exercise intensity affects both platelets and FVIII. Increases in FVIII cause post-exercise hypercoagulation. Endogenous exercise-related vasodilators such as nitric oxide (NO) affect FVIII activation and function [76]. FVIII activation may be due to circulation or release of stored or freshly synthesized FVIII [77,78]. High-circulating catecholamines promote coagulant activity through platelet response and FVIII release. Epinephrine also stimulates coagulant activity in high intravascular stress regions [44,79,80]. Catecholamines boost platelet aggregation through other agonists rather than direct platelet response (including ADP and collagen) [39,44,52,76].

##### Fibrinolysis

There is a consistent body of evidence suggesting that acute endurance exercise immediately leads to an increase in fibrinolysis [44,72,80,81]. This impact may be largely explained by an increased rate of tPA release from vascular endothelial cells [82]. Exercise is associated with a decrease in renal blood flow, which in turn leads to a drop in PAI-1 activity and a lesser clearance of tPA by the liver [44,83]. Additional predictable reactions to acute exercise include a reduction in PAI-1 activity, which is the primary inhibitor of tPA function, and an increase in fibrin degradation products (FDPs), which may include D-dimers (a byproduct of fibrin formation) [44,72,84].

#### 2.1.2. Patient Populations

##### Platelet Function

The symptom-limited cardiopulmonary exercise test (CPET) performed in a cycle ergometer has been shown to increase platelet aggregation in individuals with CAD despite antiplatelet therapy. The increased aggregation in this study was the result of not only platelet activation, but also of platelet count augmentation, indicating that an antiplatelet regimen cannot suppress platelet aggregation following exercise test until exhaustion. In this study, individuals with reduced ejection fraction receiving anticoagulants were excluded, while platelet expression levels of glycoprotein (PAC-1) and P-selectin (CD62p) remained unchanged [85]. Despite these findings and previous knowledge about the thrombotic risk in individuals with CVD in maximal exercise, a recent systematic review by Mo et al. [86], concluded that in most cases, symptom-limited CPET does not significantly affect platelet aggregation, nor serum levels of β-thromboglobulin (βTG) in individuals with CAD. Therefore, while CPET can be considered safe for individuals with CAD concerning coagulatory and fibrinolytic response, during testing a trained physician should be present as heart emergencies, hypoxemia, and vasovagal/orthostatic syncope might occur.

In 2018, Hvas and Neergaard-Petersen [42] reviewed the literature about the influence of acute exercise in platelet function in patients with CVD (CAD, angina pectoris, hypertension, and peripheral artery disease). Some studies indicated increased platelet aggregation and/or activation following exercise, whereas others reported a decrease relative to controls. During or after activity, the antithrombotic effects of aspirin are attenuated, indicating that the protection of aspirin might be limited only to the resting state. Importantly, most research found that CVD patients did not vary from healthy controls in platelet aggregation before exercise; however, some data indicated increased platelet activation in patients compared with controls.

##### Coagulation and Fibrinolysis

In patients with invasive breast cancer stage I/II, a long-term exercise program showed only a limited impact on most hemostatic factors, compared to the control group; however, it significantly increased factor VII antigen and prevented increasing the concentration of vWF following breast cancer therapy in postmenopausal women [87]. Similarly, numerous strenuous exercise-induced cardiovascular risk factors in hypertensive patients have been reported. These include endothelial dysfunction; increased thrombin generation, with elevated levels of PT, F1 + F2, and TAT; increased platelet activation and aggregation with hyperactive catecholamine response; elevated fibrinogen levels; increased plasma viscosity; abnormalities in clotting activity; altered fibrinolytic activity with increased levels of PAI-1, TAFI, and plasmin-α2 antiplasmin complex (PAP); decreased tPA activity; and prolonged euglobulin clotting lysis time [88,89]. Asymptomatic patients with moderate-to-severe aortic valve stenosis demonstrated prothrombotic alterations in coagulation and fibrinolysis markers during exercise stress testing, reacting to exercise with more prominent and extended thrombin production and impaired fibrinolysis. The authors of this study concluded that exercise-induced prothrombotic states may contribute to progression of the disease [90].

### 2.2. Chronic Adaptations

#### 2.2.1. Healthy Individuals and Athletes

##### Platelet Function

Regular endurance training improves endothelial function [91]. Endurance training has a clinically significant effect on lowering systolic blood pressure (SBP) [92]. Physically active subjects demonstrate significantly less platelet activation and hyperreactivity compared to sedentary controls [93]. A randomized controlled study found a small but significant prolongation of aPTT at rest and after acute exercise changes following 12 weeks of moderate aerobic endurance training [47]. For instance, adhesiveness and aggregation of platelets during acute strenuous exercise are attenuated with moderate-intensity training, whereas there is no change with low-intensity endurance training [94,95] Platelet count at rest seems to be lower in trained persons as well, probably because of training-related plasma volume expansion and, as a result, decreased contact time of platelets to the artery wall in trained individuals [95,96]. In addition, research has revealed that trained populations have lower levels of exercise-induced platelet activation compared to groups who have not been trained. Individuals who have undergone training have higher levels of total antioxidant stress, which results in reduced platelet reactivity to ADP and collagen [96]. Finally, decreased catecholamine release as a result of training causes less activation of platelets, along with decreased platelet sensitivity, density, and affinity of platelet surface alpha2-adrenoreceptors to plasma epinephrine. These adaptations occur as a result of repeated catecholamine exposure [82,97].

##### Coagulation

It has been shown in several studies that aPTT, PT, TT, FVII, and FVIII activity at rest and after exercise are unaffected by long-term endurance training programs, with no difference between sedentary and athletic populations [46,98,99]. Temporary activation of coagulation following acute maximum exercise, including the usual exercise-induced rise in FVIII, remained unaffected after 12 weeks of submaximal endurance aerobic training, with no post-training alterations in hemostasis [98]. Another study comparing competitive cyclists and sedentary, healthy volunteers revealed no changes in FVIII and vWF [99]. Regular exercise and training may improve hemostatic efficiency and reduce acute thrombosis by increasing endothelial sensitivity and vascular tone [47]. Regular exercise alters endothelial cells, which monitor hemostasis. Only one week of endurance training increases endothelial-dependent arterial vasodilation, reducing intravascular shear stress and platelet–vessel wall contact [100]. Increased bioavailability of NO counteracts platelet adhesion and aggregation [101]. Regular exercise increases NO and other platelet aggregation inhibitors, reducing thrombosis risk [102].

##### Fibrinolysis

An increase in fibrinolytic activity after maximum exercise in a trained group, but no change following low-intensity treadmill training was found by Ferguson and Mason Guest [103]. This study’s results on the effects of training on fibrinolysis and related markers are inconclusive; however, frequent exercise was linked to a greater fibrinolytic response to a bout of maximal acute exercise and a higher fibrinolytic activity at rest [104]. According to the findings of cross-sectional research studies, the fibrinolytic markers tPA and PAI-1 respond favorably at rest and during exercise [105,106]. Several weeks of aerobic endurance training have been shown to decrease resting fibrinolytic activity and increase the fibrinolytic response to acute exercise [39,82]. Increased resting fibrinolytic activity in trained individuals may be attributed to increased tPA release, decreased PAI-1 activity, and decreased tPA/PAI-1 complex formation at rest [44]. Another study comparing untrained and aerobically fit men revealed comparable post-exercise increases in tPA activity as well as substantial resting tPA elevations [104]. This research also revealed a decrease in resting PAI-1 activity and tPA/PAI-1 complex formation.

#### 2.2.2. Patient Populations

##### Platelet Function

Platelet reactivity is correlated with the incidence of acute thrombosis and the long-term pathophysiology of thrombosis and CVD [44,62]. Training consistently reduces exercise-induced adhesion and aggregation activity of platelets, both at rest after intense acute exercise, in populations of varying health conditions [94]. It is probable that suppressing platelet activity via regular exercise and training lowers the incidence of thrombosis at rest and during physical effort by a substantial amount [39]. Following only a few weeks of deconditioning, adaptations to platelet activity revert to their pre-training form [39,56]. Moreover, regular MICT enhances platelet mitochondrial bioenergetics in stroke patients [107] and individuals with peripheral arterial disease [108].

##### Coagulation

Hypercoagulable state and VTE are common complications in various health conditions. Active cancer and anticancer therapy are closely associated to VTE [109,110]. The incidence rate of cancer patients suffering VTE is estimated between 4 and 20% [111,112]. VTE development is connected to patient’s characteristics, the type of cancer, and pharmaceutical [113] or surgical therapy [114]. Several factors may explain the link between conventional chemotherapies or targeted therapies and thrombotic events. Although the physiological mechanism by which platinum-based agents cause a prothrombotic state is still unknown, cytotoxic medicines seem to cause endothelial damage, activation of coagulation, and platelet activation, which contribute to thrombosis [110]. Exercise prescription during treatment in patients with peritoneal carcinomatosis appears preventive in developing VTE, even when of low intensity [115]. However, existing evidence supports a stronger effect on the hematopoietic system with moderate-intensity exercise when it comes to acute exercise, while moderate or moderate-to-high intensity is recommended for a progressive exercise program [38,116]. This promotes fibrinolysis by increasing tissue levels of tPA and urokinase-type plasminogen activator (uPA) while decreasing fibrinogen and PAI-1. Furthermore, following specific exercise protocols has been shown to decrease surrogate markers of thrombin and endogenous thrombin potential (ETP) [72]. It is uncertain whether the same hematologic adaptations are observed in oncology patients who are about to receive, are already undergoing anticancer therapy, or even being prepared for surgery.

The same applies to other patient populations. More specifically, hip fracture patients have an increased risk of perioperative coagulopathy, and advanced viscoelastic methods such as rotational thromboelastometry (ROTEM) can detect early-stage high coagulation activity [117,118]. Future studies can investigate the short-term effects of preoperative endurance exercise on hemostatic balance in hip fracture patients (ClinicalTrials.gov (accessed on 31 December 2022): NCT05389800 [119]).

Coagulopathy is linked to mortality in COVID-19 patients, and D-dimer levels, in combined with poor vascular health, may increase the risk of coagulopathy and death. Low-to-moderate-intensity exercise can improve coagulopathy biomarkers, but high-intensity exercise increases thrombotic risk. Based on thromboprotective potential of endurance training in patients with cardiometabolic diseases [120], low-to-moderate-intensity exercise might be an adjuvant treatment for mild-to-moderate COVID-19 and minimize the chance of acquiring severe symptoms associated with increased mortality [121].

In elderly and cardiac patient groups, training-induced decreases in coagulation markers are more evident at rest and during acute exercise [72]. In post-myocardial patients, 4 weeks of training seemed to diminish resting FVIII levels, while fibrinogen levels dropped after a few months of aerobic exercise training [72,122]. Other studies suggest that training may reduce plasma fibrinogen concentration in elderly males, but not in young ones [123], and have found significant improvements in hemostatic markers including aPTT, FVII, prothrombin fragments 1 + 2 (F1 + F2), and vWF in post-myocardial patients and subjects between 50 and 75 years old, but not in younger healthy populations [59,124]. Normal aging and illness cause vascular changes. Endurance training may modify these characteristics, especially in unwell and older untrained populations [47]. Healthy, well-preserved endothelium adapts minimally to exercise [59,125]. In younger, healthy individuals, training may have little to no effect on FVIII and other endothelial-derived coagulation factors.

##### Fibrinolysis

Exercise’s cardioprotective benefits are partly due to a reduced tendency to produce clots and an increased fibrinolytic potential [21]. Additionally, it has been hypothesized that a person’s maximum aerobic capacity could be connected to their fibrinolytic activity levels [126]. According to the findings of many studies, the rate of acceleration of fibrinolytic activity is directly connected to the amount of work that is accomplished [81,127]. These findings imply that greater aerobic fitness may result in bigger increases in fibrinolysis when maximum exercise is performed [44,46,126,128]. The beneficial fibrinolytic adaptations that occur from exercise may be undone in a matter of weeks if the training is stopped for long enough [39]. In order to preserve hemostatic equilibrium in healthy persons, an elevated level of fibrinolysis functions as a counterbalance to the unfavorable impact of hypercoagulation [104]. This may be of pathophysiological significance in terms of lowering the susceptibility to life-threatening thrombogenic events, which may occur as a result of intensive physical activity.

Table 1 summarizes the platelet function, and the coagulatory and fibrinolytic responses in acute exercise and chronic endurance training.

## 3. Resistance Exercise and Hemostatic Balance

### 3.1. Acute Effects

Resistance exercise causes immediate alterations in all cardiovascular system components. A bout of resistance exercise affects both fibrinolytic (plasma D-Dimers, tPA, PAI-1) and coagulatory potential (plasma fibrinogen, PT, aPTT, and βTG). Main changes are observed in fibrinolytic factors. Despite a paucity of evidence in patients with various conditions, such as chronic kidney disease [129,130,131], spinal cord injury [132,133,134], or chemotherapy-induced peripheral neuropathy [135], current findings indicate that it may be a relatively safe exercise modality, with or without blood flow restriction (BFR) application, even for patients with CVD [136,137,138].

#### 3.1.1. Healthy Individuals and Athletes

##### Platelet Function

Resistance exercise affects platelet function and count differently among resistance-trained and sedentary individuals. In a study measuring platelet activation indicators after an acute heavy resistance exercise test in trained and untrained individuals, Creighton et al. [139] found that resistance-trained individuals had lower plasma βTG, an indicator of platelet activation, compared to untrained. However, from platelet surface receptor concentrations (von Willebrand factor antigen (vWF: Ag), platelet factor 4, and βTG) and platelet count, only βTG revealed significant group differences. Increased levels of βTG, along with platelet aggregation, have also been reported in other studies as an acute response to resistance training [140,141]. This increase in βTG is independent of the time-of-day effect (morning or evening) [142]. Moreover, isometric exercise does not seem to affect either platelet number [122] or platelet function [143].

Exercise increases platelet count in many ways [144]. First, the change in platelet count may be due to a brief reduction in plasma volume during and after exercise, causing hemoconcentration [145,146]. These dynamic plasma volume fluctuations may also explain why platelet count drops fast after exercise, matching the recovery of plasma volume [140]. Exercise-induced elevations in circulating catecholamines may sensitize platelets and explain platelet increases and reductions during and after exercise [140]. During exertion, the lungs, spleen, and bone marrow produce catecholamine-derived platelets [147,148]. The intensity and volume of resistance training affect catecholamine release. Resistance exercise and catecholamine release affect platelet number and activation. Increased circulating catecholamines have been suggested as the mechanism by which platelets are activated during high-intensity exercise, but studies measuring catecholamines or blocked 2-adrenoreceptor sites have shown that platelet hyperaggregation cannot be attributed exclusively to the adrenoreceptor pathway. Exercise may also enhance hypercoagulability by increasing H+ ions, hemoconcentration, and reticular tissue platelet release [62].

##### Coagulation

Training with BFR has gained ground over the past years [149,150]. This training approach seems to induce an acute hemodynamic response [151] and hemostatic disturbance [152]. Particularly, an acute bout of low-load resistance stimulus combined with BFR significantly increases fibrinolytic potential through increased tPA activity, without affecting coagulatory factors, such as PAI-1, PT, or levels of fibrinogen [152]. However, BFR should be used with caution in high-risk individuals who have undergone orthopedic surgery, due to their increased risk for VTE [153].

##### Fibrinolysis

In young women, resistance training increases fibrinolytic potential, and this change may be modulated by body composition and fat distribution, with greater increase of tPA and decrease of PAI-1 in the low-fat group. Additionally, there was tendency of post-exercise lower concentrations of TAT, a marker of net activation of coagulation [154]. These results enhance the association of metabolic syndrome with a prothrombotic state [155].

#### 3.1.2. Patient Populations

##### Platelet Function

Although platelet activation markers in patients with T2DM were found to be similar in resistance exercise with and without BFR, the former might cause thrombocytosis and increase the risk of developing blood clots [156].

##### Fibrinolysis

Acute resistance exercise has revealed similar results in males with CAD, as with healthy individuals. More specifically, tPA and PAI-1 were increased and decreased, respectively, immediately after a bout of a resistance exercise program, without elevating thrombotic potential [136].

### 3.2. Chronic Adaptations

Resistance training lowers blood pressure (BP) [157] and improves endothelial function in both healthy individuals and those with CVD or metabolic diseases [158]. Positive vascular alterations may explain why resistance training improves health [6,144]. Both acute bout and regular resistance training lead to an enhanced fibrinolytic potential, with unclear effects in coagulatory factors [84,159,160]. The primary physiological processes that explain this influence include a simultaneous PAI-1 inhibition, an increase in tPA, and a shortening of the activated partial thromboplastin time (aPTT), resulting in reduced blood clot dissolution [72,84].

#### 3.2.1. Healthy Individuals and Athletes

##### Coagulation

In a study evaluating the effect of moderate-to-heavy resistance training on several coagulation factors, fibrinogen, factor VII, and factor VIII were not affected after an 8-week training program [161]. Although the intervention program was sufficient to induce changes in strength and body composition, it had no significant impact in plasma levels of fibrinogen, coagulation factor VII, or factor VIII. Although the mechanism behind the rise in factor VIII with exercise is unknown, the β-adrenergic receptor pathway is plausibly related to the post-exercise increase of this factor. Moreover, resistance-trained individuals present a lower capacity to form a clot, as reflected by aPTT [84].

##### Fibrinolysis

Recently, Nagelkirk et al. [160] examined coagulation and fibrinolytic potential before and after an 8-week, whole-body, resistance training regimen. Plasma concentration of PAI-1 activity, PAI-1 antigen, tPA activity, and tPA antigen were used as fibrinolytic variables. In another study, Kupchak et al. [84] investigated the beneficial effects of habitual resistance training in the acute response of an acute exhaustive resistance exercise test. Compared to sedentary subjects, a greater fibrinolytic potential in the resistance-trained group was concluded. Although both groups demonstrated a significant post-exercise increase in tPA activity and a decrease in PAI-1 activity, trained subjects presented a greater increase in tPA activity [84]. Similarly, in 2007, Baynard et al. [162] evaluated fibrinolytic reactions to a maximum treadmill exercise test in endurance-trained, resistance-trained, and untrained individuals. There were no group differences in resting tPA or PAI-1 activity. The maximum treadmill test increased tPA activity, but there was no significant difference between groups. PAI-1 activity declined after maximum exercise in all groups, although less so in the resistance-trained group (70% in the resistance-trained vs. 86% and 82% in endurance-trained and untrained groups, respectively). These studies imply that resistance-trained individuals have a clear benefit of fibrinolytic potential.

Previous studies have shown regular exercise may improve fibrinolytic activity. Increased endothelial sensitivity to tPA release [83] or slower hepatic tPA clearance during exercise [163] might warrant enhanced fibrinolytic potential. Others hypothesize that frequent exercise may diminish resting PAI-1 activity [164]. Kupchak et al. [84] indicated that frequent resistance training may reduce PAI-1’s inhibitory effect and boost tPA activity, providing a favorable fibrinolytic condition.

#### 3.2.2. Patient Populations

##### Coagulation

The long-term effect of resistance training with BFR has been examined in patients with CAD. In a pilot randomized controlled trial, an eight-week program led to no changes in post-exercise levels of fibrinogen and D-dimers, despite a lowering in SBP. However, this study may indicate the safety of using BFR in patients with CAD [137,138], although larger studies are necessary to conclude on the safety with more certainty.

Table 2 summarizes the platelet function, and coagulatory and fibrinolytic responses in acute exercise and chronic resistance training.

## 4. High-Intensity Interval Exercise and Hemostatic Balance

Although a bout of vigorous exercise (≥80% $\dot{V}O$_2_max) has long been considered an unsafe option for high-risk populations due to increased platelet reactivity and coagulation [39], Bogdanis et al. [165] recently found that there is a duration effect on the amount of hematological influence. Regarding hemostatic activity, it is widely considered that blood fibrinolysis is increased beyond 68% $\dot{V}O$_2_max without activation of blood coagulation mechanisms; however, simultaneous activation of fibrinolysis and hemostatic mechanisms does not occur until 83% $\dot{V}O$_2_max (Figure 1) [39,127]. The early activation of fibrinolysis relative to hemostasis is hypothesized to be an adaptive and protective mechanism against excessive bleeding that may occur under stress [72,81,127].

HIIE has been characterized as the “best bang for the buck for time unit” exercise in terms of life expectancy and risk of CVD mortality [166]. Concerns over the safety and practicability of this training method inspired the development of a more acceptable type of HIIT, known as low-volume HIIT [167,168]. This variant entails lowering the absolute intensity of the intervals, increasing the length of the intervals, and decreasing the recovery time between intervals [167]. Different studies have shown that low-volume HIIT is a strong stimulant for morphological, metabolic, and biological enhancements in multiple physiological systems [167]. To date, there is a paucity of evidence in the safety and efficacy of a single bout of HIIE and regular HIIT in hemostatic balance, mainly for CVD and other health conditions. Interestingly, current evidence suggests an adaptive effect in repeated habitual HIIT, resulting in a less sensitive coagulatory response [46].

### 4.1. Acute Effects

#### 4.1.1. Healthy Individuals and Athletes

##### Platelet Function

Hematologic responses such as plasma volume and blood count (leukocyte, neutrophil, lymphocyte, monocyte, and platelet count) have been investigated, showing that HIIE with longer duration bouts results in a greater post-exercise change in plasma volume, leukocyte, and platelet count [165]. The same tendency has been found in platelet count in healthy adults in response to firefighting drills [169], and in elite karate athletes [170].

Platelet aggregation increases by vWF binding during intense exercise (80% $\dot{V}O$_2_max), but moderate exercise (60% $\dot{V}O$_2_max) suppresses agonist- and shear-induced platelet aggregation through decreased vWF binding to platelets [39]. It has been found that 30 min of exercise on a bicycle ergometer at around 55% $\dot{V}O$_2_max had no impact on platelet reactivity or dynamic coagulation as determined by hemostatometry [171].

##### Coagulation

In a recent study, Sackett et al. [172] evaluated 17 healthy males who performed repeated bouts of a standard Wingate anaerobic test on a cycle-ergometer of 30 s maximal work followed by 4.5 min of recovery (HIIT ratio 1:9) for eight weeks. Based on this study, there was no impact on clotting times of aPTT or prothrombin time (PT), nor on plasma concentration of TAT. Interestingly, although fibrinogen decreased significantly in the short-term (four weeks) and increased significantly between four and eight weeks, at the eight-week time point there was no significant difference compared to baseline values [172]. In contrast to Sackett et al. [172], aPTT was decreased on average by more than three seconds (9.0% and 7.7% for females and males), indicating a procoagulatory state for both males and females following firefighting activities [169]. Although there is limited research on women, recent findings indicate no sex-dependent differences in coagulatory response to high-demand activities [169]. However, a recent cross-over study demonstrated distinct sex-specific responses in a 3-month endurance and resistance training plan regarding cardiovascular risk factors [173].

In elite karate athletes, Karampour and Gaeini [170] found considerable differences between HIIE and resistance exercise. Although both types of exercise significantly reduced PT and aPTT, HIIE increased factor VIII and decreased fibrinogen more than resistance exercise. This study indicates that HIIE, via reducing the anticoagulant process, generates optimum hemostatic balance by increasing coagulation and fortifying fibrinolysis [170]. Increased perfusion of active muscle capillary beds and blood flow redistribution reduces plasma volume during exercise without a proportionate loss of plasma proteins. Increased hydrostatic and artery pressure from muscle contractions during strenuous exercise reduces interstitial fluid, whereas sweat and other fluid losses lower plasma volume [174]. Reduced plasma volume during exercise brings clotting factors closer together and exposes the endothelium to prothrombotic chemicals. When relative concentrations of these molecules rise, clotting factors may work more on each other and activate platelets, accelerating the PTT [175]. These actions may cause thrombosis, particularly if coagulant to fibrinolytic activity is high [59]. Strenuous and HIIE increase coagulatory potential.

##### Fibrinolysis

Changes in fibrinolysis generated by exercise seem to be mostly dependent on exercise intensity and duration. Exercise of higher intensity and shorter duration leads in a larger fibrinolytic reaction than exercise of moderate intensity and longer duration [44]. In addition, large changes in fibrinolysis do not occur until exercise intensity reaches at least 70% $\dot{V}O$_2_max [81,127]. Likewise, exercise performed above lactate threshold induces stronger fibrinolytic reactions than equicaloric exercise below the lactate threshold [176]. Numerous studies demonstrate that increases in fibrinolysis through tPA activation precede changes in coagulant activity [177,178], especially during vigorous exercise, when activation of fibrinolysis occurs at a lower intensity and to a higher degree than activation of coagulation [72]. This may be an adaptive reaction designed to mitigate the possible adverse consequences of increased fibrin deposition during exercise [62]. Changes in fibrinolysis caused by exercise are transient, with the return to baseline occurring between 60 min and 24 h following intense activity [179]. In a study by Smith et al. [169], multiple bouts of firefighting activity led to increased fibrinolysis (tPA antigen, tPA activity, PAI-1 antigen, PAI-1 activity) in both healthy adult males and females.

#### 4.1.2. Patient Populations

##### Platelet Function

In the majority of research, the activating effects of exercise are shown to be more prominent in populations that are sedentary in comparison to populations that are routinely active, and in patient groups in comparison to healthy control groups [72]. Platelet hyperactivation was shown to be induced by vigorous activity in males with a sedentary lifestyle, but there were no significant alterations identified in those who regularly exercised [93]. When exercising vigorously, plasma volume decreases, which generates a temporary procoagulant environment. This occurs because the clotting factors are brought physically closer together, and the endothelium is exposed to prothrombotic molecules. Clotting factors have a larger chance to act on each other and on activated platelets when there is a rise in the relative concentrations of these molecules. This may result in a shortened time required for the aPTT [175].

##### Coagulation

The relationship between acute vigorous exercise and thrombotic events has led to speculation that an elevation of coagulant activity is one plausible explanation for initiating acute coronary events. This speculation is based on the fact that acute vigorous exercise is linked to thrombotic events. The magnitude of sympathoadrenal activation on the endothelium can induce exertion-related coagulopathy, which may facilitate the thrombotic risk that occurs due to acute vigorous exercise. This risk may be facilitated by hyperactive and related responses of the sympathetic nervous system and hemostatic systems [79]. However, it is important to highlight that due to the enhanced fibrinolysis and increased blood flow, there is a minimal risk of clot formation during exercise. Yet, upon cessation of activity, cardiac output recovers to resting values. While coagulation potential stays increased for hours after cessation of activity, fibrinolytic potential drops swiftly [180,181]. Hence, the thrombotic risk might be more pronounced after activity cessation than during the actual effort.

### 4.2. Chronic Adaptations

While it is recognized that acute exercise causes hypercoagulability, regular exposure to high-intensity training may actually shape the acute hemostatic response to exercise [46]. Indeed, elite athletes and regularly high-intensity-trained individuals may be more resistant to the prothrombotic effect of vigorous exercise. Existing evidence and proposed physiological mechanisms are promising for a potential cardioprotective effect along with enhanced fibrinolytic potential in high-risk patients participating regularly in HIIT programs.

#### 4.2.1. Healthy Individuals and Athletes

##### Platelet Function

Platelet count, adhesion, and aggregation at rest and in response to acute strenuous activity are all reduced with exercise training in populations of diverse health status [54]. This effect is seen in both healthy and unhealthy individuals. This impact is thought to be caused by an increase in plasma volume, which is a pattern that has been seen in athletes who have undergone training [94,182]. It is stated that the extent of the rise in plasma volume with training depends on the intensity of the exercise. This is because higher acute losses of plasma volume induce a faster rate of hemodilution [183].

##### Coagulation and Fibrinolysis

Regular exercise reduces resting and exercise-induced catecholamine spikes [39]. One week of intense exercise reduced the norepinephrine response to a given workload. Catecholamines affect procoagulant factors differently [184]. Training lowers the density and affinity of platelet surface adrenergic receptors and exercise-related catecholamine release, resulting in decreased platelet activity and aggregation under high shear flow [52,106]. Endothelial cell sensitivity increases with exercise, and up-regulation of endothelial β2-adrenoreceptors accelerates FVIII and vWF release [63], which may explain the continued procoagulant phenotype after exercise. Exercise may adjust fibrinolytic activity via multiple mechanisms. Most popular hypotheses include endothelial sensitivity to tPA release and decreased tPA/PAI-1 formation [104,185]. Reduced hepatic tPA clearance and lipid alterations promote decreased resting PAI-1 activity and tPA/PAI-1 complex formation [44,83,163]. Training-related changes in catecholamines may be a mechanism for altered fibrinolytic activity, but data are equivocal regarding the magnitude of the effects plasma epinephrine and norepinephrine have on fibrinolysis during vigorous exercise. Therefore, the enhanced endothelial cell sensitivity for tPA release is most likely related to other non-adrenergic mechanisms.

Regular exercise and training may reduce thrombosis by desensitizing platelet α2-adrenoreceptors to exercise-induced catecholamines [63,82]. Improved hemostatic balance may be due to hormonal balance, hemodynamics, and oxidative stress [105]. Elevated antioxidant status was identified after just nine HIIT sessions, suggesting a good impact on vasomotor function intravascular stress, perhaps reducing contact-dependent activation of factors and other flow-related activators that induce thrombosis [106]. HΙΙT may reduce this risk by enhancing catecholamine responses and hemodynamics. Studies directly correlating hemostasis and sympathetic nervous system stress responses to exercise are sparse. Both contribute to hemostatic balance. Given their clinical significance on vascular health and thrombosis, their combined influence on endothelial cell activation is of special relevance. High-intensity exercise lowered PAI-1 activity, whereas low-intensity training did not [62].

#### 4.2.2. Patient Populations

##### Platelet Function

Following physical exertion, platelets have been firmly related to CVD [186,187]. While there are limited data on exercise training and platelet response, the beneficial effects generally promote a decrease in platelet activation and aggregation by down-regulating adhesion molecule activation and expression, decreasing platelet α2-adrenoreceptor performance and sensitivity to hyperaggregating effects of catecholamines, reducing platelet-vWF interaction, and improving endothelial NO release and bioavailability, which collectively support a reduction in platelet activation and aggregation [106,188]. In addition, training may reduce the contact time of platelets with the artery wall in places of high stress, making endothelial cells less susceptible to activation and subsequently enhancing hemostatic action [54]. In a randomized controlled trial, Heber et al. [189] compared the platelet function response on a 12-week HIIT plus MICT versus MICT alone in male patients with acute coronary syndrome undergoing dual antiplatelet therapy. The combined training program, HIIT plus MICT, proved more effective in reducing platelet reactivity. Moreover, platelet hyperreactivity in response to strenuous exercise was demonstrated despite receiving an antiplatelet regimen [189]. In another recent large randomized controlled study [190], a 12-week HIIT program in patients with CAD revealed no significant differences in platelet aggregation between the exercise and control group.

##### Coagulation

Intense physical activity stimulates the sympathetic nervous system and augments the hypercoagulable response principally via stimulating adrenoreceptors on the platelet and endothelial cell surfaces [55,63]. In addition, higher plasma catecholamine levels during maximum exercise enhance oxidative stress while lowering the availability of NO, which results in a procoagulant phenotype that might trigger thrombogenesis [106]. In healthy people, a procoagulant phenotype is successfully maintained by simultaneous activation of protective fibrinolytic mechanisms, such as endothelial release of tPA, and in part by the effects of catecholamines [63]. In cases of impaired endothelial function, especially in older, sedentary individuals, or those with pre-existing CVD or atherosclerotic vessels, the procoagulant phenotype induced by strenuous exercise and elevated norepinephrine and epinephrine, outweighs the thromboprotective mechanisms of action, and increases the risk for initiating a spontaneous cardiac event [106]. During strenuous exercise, high blood flow velocity and enhanced intravascular turbulence induce modest tissue damage. The degree of vascular injury is positively correlated with the extent of hemostatic alteration, where sympathoadrenal activation has a dose-dependent influence on hemostatic activity [79]. Consequently, hypercoagulability after vigorous exercise shows that sympathetic nervous system activation and shear stress might damage and activate tissue to induce intravascular thrombosis in a synergistic manner. Research has largely concentrated on either hemostasis or sympathoadrenal activity during exercise, and their linked physiological responses to stress have received less attention. Given its association with acute thrombotic development and CVD, the hemostatic–sympathetic nervous system response to exercise stress is important.

##### Fibrinolysis

When endothelial function and thromboprotective mechanisms are impaired as a result of age or disease, the collective procoagulant effects can induce harmful thrombogenicity, either immediately after exercise cessation or late into the exercise recovery period. This is especially true in elderly patients. By facilitating the development of primary prevention strategies and treatment modalities to prevent adverse thrombotic responses during strenuous physical exertion, vigorous exercise training may be able to promote longevity by manipulating physiological hemostasis and hormonal responses. This could be attempted in order to prevent adverse thrombotic responses [191].

Table 3 summarizes the platelet function, and coagulatory and fibrinolytic responses in HIIE and HIIT.

## 5. Discussion

Acute and chronic systemic adaptations of different types of exercise have long been investigated among healthy and patient populations. In this study, we aimed to review the literature regarding the hemostatic system’s acute and chronic responses to different types of exercise. Although regular moderate-intensity endurance and resistance training may be considered the “holy grail” for its multi-systemic preventive and therapeutic effects, including reducing the risk of CVD, hypertension, and thrombotic events, existing literature is debatable regarding the role of acute high-intensity exercise bouts and HIIT in hemostatic balance. Indeed, endurance and resistance training enhance fibrinolytic potential and inhibit platelet function, but are inconclusive about coagulatory factors. While professional and well-trained athletes may be at reduced risk of these adverse events induced by physical exertion, such as adrenaline-induced platelet aggregation and platelet reactivity, healthy sedentary individuals should still follow the principle of progression in intensity and time of exercise. Contradictory evidence has emerged in the safety of high-intensity exercise in individuals with existing CVD due to exercise-induced hypercoagulability, abnormal fibrinolysis, and platelet aggregation, resulting in a prothrombotic state. Therefore, it is recommended for any individual, either healthy sedentary or with existing chronic diseases, to initiate at a low-to-moderate intensity of exercise and gradually progress to more vigorous, as no safety concern is warranted from current evidence when the principle of progression is respected.

It has long been reported that platelet function, and fibrinolytic and coagulatory responses to acute and regular exercise seem to be different among sedentary and trained individuals. The current evidence indicates a protective and inhibitory role of trained status in platelet activation [192,193,194]. In a clinical trial published in 2018, Lundberg Slingsby et al. [195] recruited 42 middle-aged, healthy, normal-weight males who were separated into three groups with different training level statuses. Compared to the untrained participants, the well-trained subjects’ platelets showed a lower basal reactivity, a greater sensitivity to the anti-aggregation effects of prostacyclin, and were more potently suppressed by dual-antiplatelet therapy. The moderately and highly trained participants showed higher vascular function in comparison to the untrained subjects, and their platelets were more inhibited by passive movement, flow-mediated dilatation, and one-leg knee-extensor exercise [195]. Similarly, Olsen et al. [38] recently reviewed the influence of acute exercise and regular training on hemostasis, illustrating the increased exercise-induced thrombogenic response in sedentary individuals compared to trained ones. Specifically, a number of compensating factors are reported in trained individuals that mitigate an exercise-induced prothrombotic state, such as diminished or reduced platelet aggregation, reduced platelet reactivity, and increased prostacyclin and NO enzyme expression [38].

Depending on the modalities of physical training, different exercise stimuli activate distinct cellular and molecular signaling pathways [196,197,198]. Thus, endurance, resistance, and interval training lead to a distinct hemostatic response. The same applies to healthy individuals compared to patients [44,62,88] (Figure 2). For example, regardless of being hypertensive or normotensive, acute endurance exercise has been found to increase tPA antigen and activity. However, PAI-1 antigen and activity are increased post-exercise only in hypertensive individuals [88]. In general, moderate-intensity exercise leads to smaller hemostatic disturbances, especially in well-trained and healthy individuals [38], in contrast to those who are hypertensive who have a slower recovery to baseline values [88]. It is worth noting that platelet aggregation in healthy individuals is not affected by very short-duration exercise [199], but by maximal efforts [50].

Recent emerging evidence indicates that epigenetics may play a role in regulating hemostatic balance [200,201]. It is estimated that most CVDs are approximately 30% heritable [202], and CADs are as high as 40–60% [203,204]. Diet, exercise, environmental pollutants, and drug use are thought to explain the remaining difference [201]. Methylation of DNA at CpG islands, histone post-translational modifications (PTMs), and microRNAs (miRNAs) are the key epigenetic regulators of gene expression. Epigenetic control is primarily dynamic and may be manipulated to affect gene expression [201]. Regular exercise also has a multi-system epigenetic effect [205,206] via hypomethylation of DNA and histone acetylation, so epigenetic modification of exercise is a possible additional benefit in the hemostatic balance. Newly discovered miRNAs modulate platelet function [207]. MiRNAs are involved in exercise-associated gene expression alterations in peripheral blood mononuclear cells, neutrophils, and skeletal muscle in non-trained and trained persons [208,209,210]. In 2011, Baggish et al. found changed circulating miRNA expression in athletes after acute and chronic exercise [211]. Exercise treatment also relieves peripheral arterial disease (PAD) symptoms and partly restores mitochondrial function [212,213]. Exercise has been found to lower miRNA-494 and miRNA-696 [214,215]. Epigenetic variation may also enable physical inactivity to affect platelet function independently or synergistically [216].

Regarding resistance training, there is less research on it than on endurance. A few mechanistic hypotheses exist on how regular resistance training can influence hemostatic balance. Epidemiological evidence suggests an association between fibrinolytic factors and hypertension [27]. An impaired fibrinolytic potential resulting from a prothrombic state in hypertensives is inversely correlated with diastolic blood pressure (DBP) [217]. Despite previous knowledge about the absence of any effect of resistance training in BP [218], there is clear evidence about resistance training efficacy in lowering BP in hypertensive individuals regardless of receiving any treatment [157,219,220,221]. BP response also is type-specific, with resistance training having a greater impact on DBP than SBP [222]. Moreover, acute hypotension following different resistance exercise protocols has been described in the literature, which lasts for up to 2 h post-exercise [223,224,225].

Another major indicator of cardiovascular health is the vascular endothelial function, which relates to the body’s capacity to maintain vascular tone homeostasis [226]. The vascular function and morphology in female elite long-distance runners are no different compared to healthy inactive females of the same age [227]. However, a young water polo Olympic team demonstrated improved arterial wall properties and endothelial function, compared with matched recreational athletes and sedentary controls [228]. Similarly, the endothelial function may be improved in individuals with prehypertension or hypertension by a variety of exercise training regimens in a manner that is clinically equivalent [222].

Although the bout duration (low- vs. high-volume), time intervals (short- vs. long-intervals), and total period performed (short-term vs. long-term) mediate in the effectiveness of HIIT protocols [229,230], the performance and health benefits might be equal when the actual workload is matched between two HIIT protocols [231]. HIIT has been shown to be a superior mode of exercise compared to MICT for improving vascular function as evaluated by flow-mediated dilation of the brachial artery (FMD) and aortic pulse wave velocity (PWV) [232,233,234,235], reducing 24 h BP [236,237,238], and arterial stiffness [239,240]. Interestingly, HIIT has been reported to provide additional benefits in platelet function when combined with MICT, compared to MICT alone, in individuals with post-acute coronary syndrome undergoing an antiplatelet regimen [189]. Clearly, vigorous exercise temporarily increases thrombotic risk; yet, chronic exposure to severe exercise stress may lower the acute stress response and reduce the likelihood of unfavorable thrombosis. Therefore, physical activity is capable of both preventing and inducing primary cardiac events and development of CVD, and these effects seem to be intensity-dependent [39,241]. Future research might focus on combining different exercise modes [242,243] and manipulating each component separately, such as bout duration [165,244] or investigating for the minimal dose needed to attain the benefits of a training program in maintaining hemostatic balance. Moreover, most advanced laboratory methods for the evaluation of hemostasis such as viscoelastic studies may be valuable in order to further clarify which part of the coagulation mechanism is most affected by different types of exercise [117,245,246,247].

## 6. Conclusions

This review presents the effects of different types of acute exercise and regular training on hemostatic balance in healthy individuals and patients with chronic diseases. The hemostatic response on a single bout of exercise is proportional to the exercise intensity, the training status, age, and having a chronic disease. A hypercoagulability state in response to acute high-intensity exercise has been described, irrespective of the health status. Vigorous bouts increase platelet function and aggregation, platelet count, factor VIII, TAT, F1 + 2, vWF, and fibrinogen and shorten the clotting time of aPTT, compared to moderate-intensity exercise. Simultaneously, an increase in fibrinolysis is intensity-dependent, although it requires at least a moderate-intensity stimulus to activate. Chronic adaptations of regular training should be investigated more. Nonetheless, such adaptations correlate with a thromboprotective effect in elite athletes and well-trained individuals. Furthermore, resistance exercise is a safe option, with less impact on the hemostatic profile. Despite limited evidence, patients with chronic diseases receive the most benefits from participating in an exercise program of any type, as their vascular health has a greater potential for improvement. Although more studies are required for clear recommendations, training programs combining moderate- and high-intensity exercise might provide the most prominent benefits in regulating hemostatic balance at rest and during exercise, especially for patients with chronic diseases.

## Figures and Tables

**Figure 1 sports-11-00074-f001:**
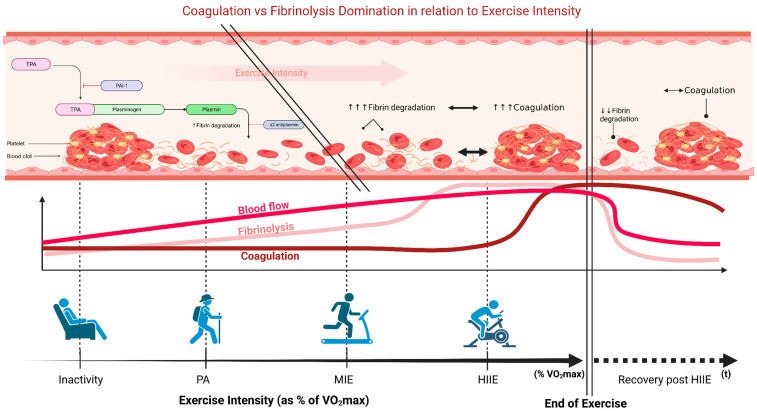
Graphic illustration of the intensity-dependent hemostatic response during a single bout of exercise. The two oblique lines mark the point of initiation of most hemostatic changes. Blood fibrinolysis is increased without activation of blood coagulation mechanisms until intensities above approximately 85% of $\dot{V}O$_2_max, where simultaneous activation of fibrinolysis and coagulation mechanisms occur. Immediately after exercise, fibrinolytic potential decreases rapidly, while coagulation potential remains elevated for hours after cessation. Adapted from ‘‘Process of Blood Clot Formation”, by BioRender.com (accessed on 13 March 2023). Retrieved from https://app.biorender.com/biorender-templates (accessed on 13 March 2023). PA, Physical Activity; MIE, Moderate Intensity Exercise; HIIE, High Intensity Interval Exercise; ↑ = increase; ↓ = decrease; ↔ = no change/no difference.

**Figure 2 sports-11-00074-f002:**
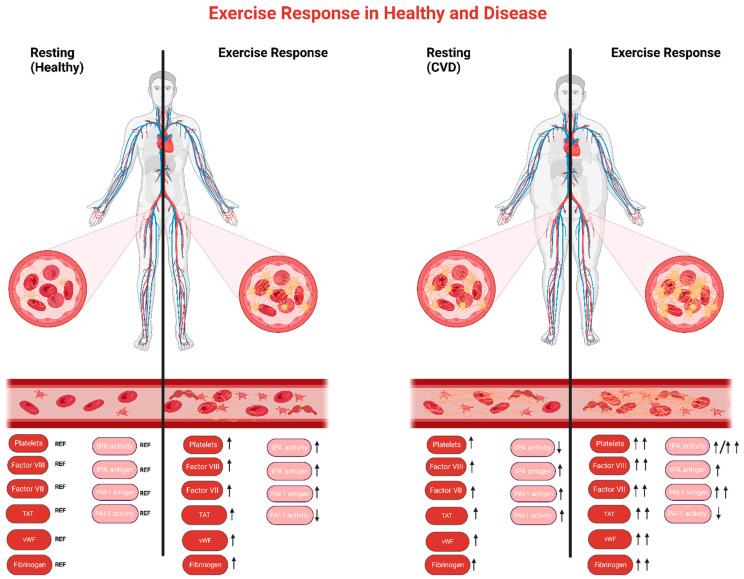
Platelet aggregation, and coagulatory and fibrinolytic responses in acute exercise. The graph shows hemostatic adaptations in response to an exercise session between healthy and CVD patients. Arrows represent changes related to reference resting values of healthy. ↑ = increase (post vs. pre and patients compared to healthy); ↓ = decrease. Red = coagulatory markers; Pink = fibrinolytic markers; vWF = von Wilebrand factor; tPA = tissue plasminogen activator; PAI-1 = plasminogen activator inhibitor-1; TAT = thrombin–antithrombin complex. Created with BioRender.com (accessed on 13 March 2023).

**Table 1 sports-11-00074-t001:** Platelet function, and coagulatory and fibrinolytic responses in acute exercise and chronic endurance training. Post- versus pre-exercise differences in absolute terms and relative differences in healthy versus patients with cardiovascular diseases are presented.

Endurance Exercise						
	Acute Response		Chronic Adaptations			
	Healthy/Patients (Post- versus Pre-Exercise Differences)	Patients vs. Healthy	Healthy/Patients (Post- versus Pre-Training Differences)		Patients vs. Healthy	
			Resting	After Exercise	Resting	After Exercise
**Platelet Function**						
Platelet aggregation and activation	↑↔	?	↓	↓		
vWF binding	↑					
Platelet count	↑					
βTG	↑↔		↓			
**Fibrinolysis**						
Clot lysis time		↑				
tPA activity	↑	↑↔	↑	↑↔	↓	↑↔
tPA antigen	↑	↔			↑	↔
PAI-1 activity	↓	↓↔	↓	↓	↑	↔
**Coagulation**						
aPTT	↓		↑↔	↓↔	↔	↔
PT	↑	↑	↔	↔	↔	↔
TT	↑		↔	↔	↔	↔
FVIII antigen	↑		↓↔	↔	↑	
vWF antigen/activity	↑		↓↔	↔	↑	↑
ETP	↑	↑	↓			
TAT	↑	↑			↑	↑
Fibrinogen	↑	↑	↓		↑	

vWF = von Wilebrand factor; βTG = β-thromboglobulin; tPA = tissue plasminogen activator; PAI-1 = plasminogen activator inhibitor-1; aPTT = activated partial thromboplastin time; PT = prothrombin time; TT = thrombin time; ETP = endogenous thrombin potential; TAT = thrombin–antithrombin complex. ↑ = increase (post vs. pre)/increased compared to healthy; ↓ = decrease (post vs. pre)/decreased compared to healthy; ↔ no change/no difference; ? = unclear.

**Table 2 sports-11-00074-t002:** Platelet function, and coagulatory and fibrinolytic responses in acute exercise and chronic resistance training. Post- versus pre-exercise differences in absolute terms and relative differences in healthy versus patients with cardiovascular diseases are presented.

Resistance Exercise						
	Acute Response		Chronic Adaptations			
	Healthy/Patients (Post- versus Pre-Exercise Differences)	Patients vs. Healthy	Healthy/Patients (Post- versus Pre-Training Differences)		Patients vs. Healthy	
			Resting	After Exercise	Resting	After Exercise
**Platelet Function**						
Platelet aggregation and activation	↑		↓			
Platelet count	↑					
βTG	↑		↓			
**Fibrinolysis**						
tPA activity	↑	↔	↑↔	↑		
tPA antigen			↔			
PAI-1 antigen			↓			
PAI-1 activity	↓↔	↔	↓↔			
**Coagulation**						
aPTT			↓	↓		
PT	↔					
FVIII antigen			↔			
vWF antigen/activity			↔			
TAT	↓					
Fibrinogen	↔		↔	↔		

vWF = von Wilebrand factor; βTG = β-thromboglobulin; tPA = tissue plasminogen activator; PAI-1 = plasminogen activator inhibitor-1; aPTT = activated partial thromboplastin time; PT = prothrombin time; TAT = thrombin–antithrombin complex. ↑ = increase (post vs. pre)/increased compared to healthy; ↓ = decrease (post vs. pre)/decreased compared to healthy; ↔ no change/no difference.

**Table 3 sports-11-00074-t003:** Platelet function, and coagulatory and fibrinolytic responses in HIIE and HIIT. Post- versus pre-exercise differences in absolute terms and relative differences in healthy versus patients with cardiovascular diseases are presented.

High-Intensity Interval Exercise						
	Acute Response		Chronic Adaptations			
	Healthy/Patients (Post- versus Pre-Exercise Differences)	Patients vs. Healthy	Healthy/Patients (Post- versus Pre-Training Differences)		Patients vs. Healthy	
			Resting	After Exercise	Resting	After Exercise
**Platelet Function**						
Platelet aggregation and activation	↑		↓↔	↓↔		↔
vWF binding			↓			
Platelet count			↓	↓		
**Fibrinolysis**						
tPA activity	↑			↑		
tPA antigen	↑					
PAI-1 activity	↓		↓	↓		
**Coagulation**						
aPTT	↓		↔			
PT	↓		↔			
FVIII antigen	↑					
TAT			↔			
Fibrinogen	↓		↔			

vWF = von Wilebrand factor; βTG = β-thromboglobulin; tPA = tissue plasminogen activator; PAI-1 = plasminogen activator inhibitor-1; aPTT = activated partial thromboplastin time; PT = prothrombin time; TAT = thrombin–antithrombin complex. ↑ = increase (post vs. pre)/increased compared to healthy; ↓ = decrease (post vs. pre)/decreased compared to healthy; ↔ no change/no difference.

## Data Availability

Not applicable.

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
