# Peer review of "The Acute and Chronic Effects of Resistance and Aerobic Exercise in Hemostatic Balance: A Brief Review"

_sports, 2023, doi:10.3390/sports11040074_

Round 1

Reviewer 1 Report

The manuscript  “The acute and chronic effects of resistance and aerobic exercise in hemostatic balance: A brief review” presents current knowledge on the effects of aerobic and resistance exercise on coagulation and fibrinolysis systems and platelet function in healthy subjects and those with chronic diseases, mainly CVD. The topic undertaken by the authors is of great practical importance mainly because of the risk of thrombotic complications in people with chronic diseases who undertake physical activity. For them, the right combination of physical exercise that favourably influences the haemostatic system can bring significant health benefits.

An overview of the articles on this topic is provided by constructive discussion at the end of the review, which attempted to elucidate the mechanisms of differential responses of the haemostatic system to exercise in healthy active subjects, healthy sedentary subjects and patients with cardiovascular disease.

No review on the field has appeared in the last five years. The cited references are mostly recent publications. Of the 217 cited references, approximately half are from the last five years. The conclusions are consistent with the evidence and arguments presented in the paper.

Comments, Concerns, and Suggestions:

·        The article lacks summary tables, so the reader can easily and quickly review current research results. I believe the inclusion of such a table (or several tables) will make the article easier to read and understand.

·        I think a little more should be written in the discussion about the molecular/epigenetic mechanisms of the effects of exercise on the haemostatic system.

·        The phrase “better arterial stiffness” (line 686) seems confusing to the reader. It may suggest that the stiffer the artery, the better, but we know it is about improved arterial wall properties and not an increase in its stiffness. I suggest correcting this phrase.

Author Response

Thank you very much for your positive and thoughtful comments.

Please see the attachment for our responses.

Reviewer 2 Report

I have read the presented review with great interest. It is a very well-written text. It covers all the main points regarding hemostatic balance and exercise. However, the manuscript is very dense and hard to follow in just one reading. I suggest that it would benefit from some tables or figures that will break the text into simplified parts.

Author Response

We are really grateful for your comments. Please see the attachment for our response to your review.

Reviewer 3 Report

General Comments:

There are several parts of the manuscript that are grammatically incorrect and/or unclear. If English is not the primary language for the authors, I would recommend a grammatical review from someone or utilize editing software. In several parts of the manuscript, the authors seem to imply that exercise safety is linked to hemostatic response. While I would agree that increased coagulation potential and/or decreased fibrinolytic potential creates an environment more conducive to an ischemic event, it is not known how much coagulation or fibrinolytic potential needs to be altered to compromise safety from a physiological perspective. It should also be kept in mind that the increased blood flow during exercise makes it extremely unlikely that an occlusive thrombus would develop during the actual exercise bout.

Specific Comments:

Page 2, lines 74-75: The author’s state that “the safety and efficacy of single-bout exercise of any type, regular resistance training and HIIT are yet under investigation”. While there may be a paucity of data on the acute risk of resistance training, there is a fair amount of literature reporting the incidence of cardiovascular events resultant from vigorous exercise. I would contend that HIIT training falls in that domain. For an example, I would suggest the following reference:

Thompson PD, Franklin BA, Balady GJ, Blair SN, Corrado D, Estes NA 3rd, Fulton JE, Gordon NF, Haskell WL, Link MS, Maron BJ, Mittleman MA, Pelliccia A, Wenger NK, Willich SN, Costa F; American Heart Association Council on Nutrition, Physical Activity, and Metabolism; American Heart Association Council on Clinical Cardiology; American College of Sports Medicine. Exercise and acute cardiovascular events placing the risks into perspective: a scientific statement from the American Heart Association Council on Nutrition, Physical Activity, and Metabolism and the Council on Clinical Cardiology. Circulation. 2007 May 1;115(17):2358-68. doi: 10.1161/CIRCULATIONAHA.107.181485. PMID: 17468391.

Page 2, line 87: The authors point out that there is “contradictory evidence on the safety of a vigorous single bout exercise”. First, this should be changed to “of exercise”. Secondly, please elaborate on the study findings cited and why these findings represent contradictory evidence. Lastly, depending on whether that statement was based on determination of hemostatic responses, I would caution basing an overall determination of exercise safety solely on hemostatic response.

Page 3, Line 100: Change “enhances also” to “also enhances”. This is just one of the grammatical issues. Please do a complete grammatical review of the manuscript.

Page 3, Line 126: Change “demonstrates also” to “also demonstrates”.

Page 3, Lines 126-127: I don’t understand the phrase “highest activity in the morning but during exercise.”

Page 4, Line 130: “performed in cycle ergometer” is not proper grammar.

Page 4, Lines 154-155: Do you mean “exercise test until exhaustion”?

Page 4, Line 161: I think it’s an overstatement to say that CPET can be considered safe based solely off hemostatic response data.

Page 4, Line 166: The sentence “During or after activity, aspirin proved less effective” is unclear and incomplete. It was less effective than what?

Page 4, Line 193: “was found a small” should be revised. “found a small” would be acceptable.

Page 4, Lines 198-199: I don’t understand how the efficacy of moderate vs low intensity training on performance is causally linked to the respective effects on platelets. That’s a very odd conclusion and needs to be changed or elaborated upon for clarification.

Page 5, Line 229: Change “frequently exercise” to “frequent exercise”. At least that appears to be the proper revision. I don’t fully understand the point made in this sentence due to the grammar.

Page 5, Lines 233-234: Please elaborate on how you are making the conclusion that exercise training decreases resting fibrinolytic activity. Was this based on decreased tPA activity? Increased PAI-1? Decreased euglobulin clot lysis time? This is somewhat of a surprising conclusion and it’s not obvious what it’s based on. References are given but please elaborate on what specific findings from those references are being used to come to that conclusion.

Page 5, Lines 234-235: Change “another research” to “another study”.

Page 7, Line 316: Again, I would suggest that using the term “safe” based on hemostatic response data is a bit of an overstatement.

Page 9, Line 418: Please elaborate on how one can conclude from that study that using BFR with resistance training is safe. 

Page 10, Lines 488-489: The phrase “aerobic capacity reaches at least 70% VO2max” is confusing. Aerobic capacity IS VO2max. Do you mean “exercise intensity reaches at least 70% VO2max?

Page 11, Lines 513-520: I agree that increased coagulation potential may influence risk of acute thrombotic events triggered by exercise. However, I think it’s important to point out that during exercise there is not much likelihood of clot formation due to the increased fibrinolysis and the increased rate of blood flow. However, once exercise stops, cardiac output returns to near resting levels. Furthermore, fibrinolytic potential decreases rapidly while coagulation potential remains elevated for hours after exercise cessation. I would point to a few key papers on this issue:

Lin X, El-Sayed MS, Waterhouse J, Reilly T. Activation and disturbance of blood haemostasis following strenuous physical exercise. Int J Sports Med. 1999 Apr;20(3):149-53. doi: 10.1055/s-2007-971109. PMID: 10333090.

Paton CM, Nagelkirk PR, Coughlin AM, Cooper JA, Davis GA, Hassouna H, Pivarnik JM, Womack CJ. Changes in von Willebrand factor and fibrinolysis following a post-exercise cool-down. Eur J Appl Physiol. 2004 Jul;92(3):328-33. doi: 10.1007/s00421-004-1098-1. Epub 2004 Apr 20. PMID: 15098129.

Page 12, Lines 612-613: The context of the “harmful thrombogenic consequences” isn’t clear to me. What is meant by “immediately”. Do you mean immediately during exercise? If so, please see my comments immediately prior. 

Page 14, Lines 675-676: I would respectfully disagree that BP doesn’t decrease as an adaptation to resistance training. I believe the following meta-analysis quite nicely shows it does reduce blood pressure.

Cornelissen VA, Smart NA. Exercise training for blood pressure: a systematic review and meta-analysis. J Am Heart Assoc. 2013 Feb 1;2(1):e004473. doi: 10.1161/JAHA.112.004473. PMID: 23525435; PMCID: PMC3603230.

Page 14, Lines 719-729: I would disagree that fibrinolytic response is not intensity dependent. In my opinion, the following demonstrate that it is intensity dependent even above the moderate intensity domain.

Womack CJ, Rasmussen JM, Vickers DG, Paton CM, Osmond PJ, Davis GL. Changes in fibrinolysis following exercise above and below lactate threshold. Thromb Res. 2006;118(2):263-8. doi: 10.1016/j.thromres.2005.06.016. Epub 2005 Aug 2. PMID: 16081145.

Andrew M, Carter C, O'Brodovich H, Heigenhauser G. Increases in factor VIII complex and fibrinolytic activity are dependent on exercise intensity. J Appl Physiol (1985). 1986 Jun;60(6):1917-22. doi: 10.1152/jappl.1986.60.6.1917. PMID: 3087936.

Davis GL, Abildgaard CF, Bernauer EM, Britton M. Fibrinolytic and hemostatic changes during and after maximal exercise in males. J Appl Physiol. 1976 Mar;40(3):287-92. doi: 10.1152/jappl.1976.40.3.287. PMID: 931838.

Author Response

Thank you for your thorough review and beneficial comments.

Round 2

Reviewer 3 Report

Lines 249-251: Increased tPA release and decreased PAI-1 activity would suggest increased fibrinolysis, not decreased.

Author Response

Dear reviewer,

Indeed, "decreased." was written by mistake there.  We corrected it to "increased" (lines 249-251).

Thank you for your suggestions.